# The Relationship between TNF-a, IL-35, VEGF and Cutaneous Microvascular Dysfunction in Young Patients with Uncomplicated Type 1 Diabetes

**DOI:** 10.3390/biomedicines11102857

**Published:** 2023-10-22

**Authors:** Jolanta Neubauer-Geryk, Melanie Wielicka, Małgorzata Myśliwiec, Katarzyna Zorena, Leszek Bieniaszewski

**Affiliations:** 1Clinical Physiology Unit, Medical Simulation Centre, Medical University of Gdańsk, 80-210 Gdansk, Poland; melanie.wielicka@gmail.com (M.W.); lbien@gumed.edu.pl (L.B.); 2Department of Pediatrics, Northwestern University Feinberg School of Medicine, Division of Neonatology, Ann Robert H. Lurie Children’s Hospital of Chicago, Chicago, IL 60611, USA; 3Department of Pediatrics, Diabetology and Endocrinology, Medical University of Gdansk, 80-211 Gdansk, Poland; malgorzata.mysliwiec@gumed.edu.pl; 4Department of Immunobiology and Environment Microbiology, Medical University of Gdańsk, 80-211 Gdańsk, Poland; kzorena@gumed.edu.pl

**Keywords:** VEGF, IL-35, TNF-α, proangiogenic cytokines, anti-inflammatory cytokines, pro-inflammatory cytokines, skin microcirculation, microangiopathy, type 1 diabetes mellitus, children

## Abstract

The aim of this study was to analyze the relationship between immunological markers and the dysfunction of cutaneous microcirculation in young patients with type 1 diabetes. The study group consisted of 46 young patients with type 1 diabetes and no associated complications. Microvascular function was assessed with the use of nail fold capillaroscopy before and after implementing post-occlusive reactive hyperemia. This evaluation was then repeated after 12 months. Patients were divided into two subgroups according to their baseline median coverage (defined as the ratio of capillary surface area to surface area of the image area), which was established during the initial exam (coverage_BASE_). Additionally, the levels of several serum biomarkers, including VEGF, TNF-a and IL-35, were assessed at the time of the initial examination. HbA1c levels obtained at baseline and after a 12-month interval were also obtained. Mean HbA1c levels obtained during the first two years of the course of the disease were also analyzed. Patients with coverage_BASE_ below 16.85% were found to have higher levels of VEGF and TNF-α, as well as higher levels of HbA1c during the first two years following diabetes diagnosis. Our results support the hypothesis that the development of diabetic complications is strongly influenced by metabolic memory and an imbalance of pro- and anti-inflammatory cytokines, regardless of achieving adequate glycemic control.

## 1. Introduction

Type 1 diabetes is associated with a dysregulation of the immune system, leading to rapid destruction of pancreatic beta cells [1]. Macrophages and T lymphocytes appear to play the most important role in this process [2].

Hypoxia is the primary stimulus for both physiological and pathological angiogenesis [3]. The process of angiogenesis, i.e., new blood vessel formation, takes place within the microcirculation and is triggered by reduced oxygen delivery. On the other hand, hyperperfusion of the capillary bed with arterial blood leads to reduced secretion of the angiogenic factor and to a process thinning of the capillary system. This is related to the abnormal activation of the Flk-1 receptor, which leads to elevated levels of VEGF to compensate for the deficiency in VEGF signaling. These high concentrations of VEGF lead to increased vascular permeability throughout the body [4]. An intriguing feature of abnormal angiogenesis is that both excessive and insufficient angiogenesis can occur in different organs in the same individual [3].

Studies have implicated that interleukin-35 (IL-35), a known anti-inflammatory cytokine in the IL-12 family, improves glycemic control and protects against DM1 by regulating the polarization of macrophages and the proportion of T cell-related cytokines [5,6]. IL-35 is not found in all human tissues, but may be induced in response to inflammatory cytokines [7]. Several studies [8,9,10,11,12] have suggested that IL-35 is a reactive anti-inflammatory cytokine that may be upregulated during acute inflammatory responses or in chronic inflammatory diseases, such as atherosclerosis. In a study by Lin et al. [8], plasma IL-35 levels in patients with acute coronary syndrome and stable angina were compared to levels in controls with chest pain. They found a positive correlation between lower plasma IL-35 levels and left ventricular ejection fraction [8].

Plasma IL-35 levels have been demonstrated to positively correlate with the severity of sepsis [9]. In a study by Du et al., the authors demonstrated that patients with septic shock had the highest serum IL-35 levels and that those levels returned to normal values with effective sepsis treatment. These results indicate that IL-35 may be used as a biomarker for sepsis. Another recent study found that IL-35 was superior to procalcitonin, C-reactive protein and other markers of early-onset sepsis [10]. Interleukin-35 inhibits the activation of vascular endothelial cells by blocking MAPK-AP1-mediated VCAM-1 expression during acute inflammation induced by lipopolysaccharides. This leads to inhibition of the acute vascular endothelial response.

Early reports have shown that ectopic expression of IL-35 by pancreatic β-cells led to significant, long-term protection against autoimmune diabetes in non-obese diabetic mice, an animal model for human type 1 diabetes [11]. Experimental studies suggest that IL-35 may have significant therapeutic potential in the treatment of DM1 and its associated cardiovascular complications [12,13].

VEGF plays a major role in the development of diabetes-related complications. In many tissues and cells, VEGF induces proliferation, migration and vasopermeability of vascular endothelial cells. It has been identified as a major trigger for the development of both nonproliferative and proliferative diabetic retinopathy. It also plays an important role in the development of neuropathy and nephropathy in diabetic patients [14]. Its activity is directly related to its expression. This means that small changes in its concentration influence physiological effects [4,15]. Some research has shown that the serum concentration of VEGF corresponds with the clinical activity of various diseases, including tumor growth, atherosclerosis of the coronary arteries, Kawasaki disease, neovascularization of synovial tissue in rheumatoid arthritis and diabetic microangiopathy [16,17,18,19]. In patients with DM1, hyperglycemia stimulates VEGF upregulation. VEGF then initiates neovascularization and increases the endothelial basement membrane thickness and vascular permeability. It also regulates the inflammatory response and endothelial apoptosis. Despite the significant body of literature describing how VEGF contributes to the development of diabetic complications, there is a lack of data analyzing the relationship between serum VEGF levels and skin microcirculation.

TNF-α is one of the main cytokines involved in the inflammatory response. It is mainly produced by monocytes and macrophages and enhances their cytotoxicity. Its biological effects depend on TNF secretion levels and intensities [20]. TNF-α is also associated with diabetic complications [21,22]. TNF-α has been found in the serum of children and young adults with type 1 diabetes and non-proliferative diabetic retinopathy [22,23]. The same authors have shown that the TNF-α level may be an independent predictor for the development of non-proliferative diabetic retinopathy in children with type 1 diabetes [22]. Similarly, higher TNF-α levels have been found in adults with type 1 diabetes and proliferative retinopathy [24,25].

Several studies have pointed out a noteworthy correlation between changes in skin microcirculation and the development of diabetic complications. When assessing microcirculation with capillaroscopy—both qualitatively and quantitatively—authors have found that changes in the exam were associated with the presence of retinopathy, nephropathy and neuropathy [26,27,28,29].

Capillaroscopy or videocapillaroscopy is a method commonly used by researchers in the assessment of cutaneous microvasculature and evaluation of endothelial function. Researchers have utilized capillaroscopy when studying various disease entities, namely peripheral vascular disease [30], hypertension [31], chronic kidney disease [32], type 1 diabetes [27,33,34,35,36,37], type 2 diabetes [38], obesity [39,40,41] or post-bariatric procedures [42]. Microcirculation reactivity has also been investigated in patients suffering from infective endocarditis [43] or ischemic, with no obstructive coronary artery disease [44].

For our study on pediatric patients, we chose capillaroscopy with employed non-selective stimuli, such as the post-occlusive reactive hyperemia test. Reactivity tests, such as the venous occlusion test and arterial post-occlusive reactive hyperemia (PORH), have been suggested to improve capillary recruitment. They allow for the determination of the total maximal capillary density with high reproducibility [45]. PORH refers to the increase from baseline in cutaneous blood flow following a brief period of arterial occlusion [46]. A wide range of brachial artery occlusion times has been described, with studies implementing times anywhere from 1 min to 15 min. A positive correlation has been described between the post-occlusive hyperemic response and the duration of arterial occlusion [46]. PORH is a widely used test for the evaluation of microcirculation, with the main underlying mechanism being shear stress and its effects [47]. Microcirculatory function, specifically the degree of capillary recruitment during occlude reactive congestion, is associated with endothelium-dependent vasodilation at the pre-capillary level. This is mediated by an axonal reflex [48,49] and endothelium-derived hyper-polarizing factor. The PORH test induces relaxation of the vascular muscle [50] and the release of local mediators and metabolites from the ischemic tissue [51]. It has also been demonstrated that local mediators, in particular large conductance calcium-activated potassium channels [49], play an important role in the PORH mechanism [48].

Capillaroscopic images often correspond with immunological findings in connective tissue diseases as well. However, little is known about the relationship between immune biomarker activity and microvascular function [52,53,54,55,56,57,58] and the reactivity of cutaneous microcirculation in young patients with type 1 diabetes. Therefore, we aimed to analyze the relationship between serum levels of immunological markers and cutaneous microcirculatory dysfunction in young patients with type 1 diabetes.

## 2. Materials and Methods

### 2.1. The Study Design and Population

The study group consisted of 46 patients with type 1 diabetes (24 girls and 22 boys, with an average age of 15.3 ± 2.4 years) (Table 1). We included patients with a minimum diabetes duration of 1.2 years. Patients were recruited from the Department of Pediatrics, Diabetology and Endocrinology at the Medical University of Gdansk between 2014 and 2018. Based on medical history, physical examination and biochemical analysis, none of the subjects had any form of microangiopathy, including retinopathy, nephropathy or neuropathy. The absence of microangiopathy (retinopathy, nephropathy and neuropathy) was confirmed using previously published criteria [59,60].

HbA1c levels during the first two years of the disease duration were obtained from the medical history of patients from the Department of Pediatrics, Diabetology and Endocrinology.

Patients in all study groups were euthyroid at the time of the study. All examinations were conducted between 8 a.m. and 1 p.m. The study protocol included medical history, capillaroscopy and laboratory tests.

The patients were examined on two occasions with an interval of 12 months between examinations. The cytokine test was only carried out during the initial examination.

All procedures performed in studies involving human subjects were in accordance with the ethical standards of the Ethics Committee of Gdansk and the 1964 Declaration of Helsinki and its subsequent amendments or comparable ethical standards. The study protocol was approved by the Medical Ethics Committee at the Medical University of Gdansk (NKBBN/277/2014; NKBBN/277 -512/2016). By entering the study, each participant provided informed consent. Parents consented to the microcirculation study and were present during the study.

### 2.2. Evaluation of Cutaneous Microcirculation

As previously described, skin microcirculation was evaluated by capillaroscopy with the use of dedicated software [33]. Study participants were instructed not to perform manicures for 2 weeks prior to the study. During this study, patients remained in a comfortable sitting position with their arm comfortably supported and their hand placed under the capillaroscope. Body temperature was monitored with a contactless thermometer (Novama—NT model) and was within the normal range in all study patients. Room temperature was controlled by air conditioning and was maintained at the same level during all tests.

Images were taken using a digital camera (5MPx; OPTA-TECH, Warsaw, Poland) connected to a capillaroscope (OPTATECH, CS-CREATIVE SOLUTIONS Group, Warsaw, Poland) and archived on a disk. Image analysis was used to determine the ratio of the area of visible capillaries to the total area of the analyzed image (coverage) [33,61].

To assess the reactivity of skin microcirculation, a post-occlusion reactive hyperemia (PORH) test was completed after a 20 min rest period. A blood pressure cuff was placed around the arm of the patient’s non-dominant limb and inflated to a pressure 50 mm Hg higher than the systemic systolic pressure to stop blood flow completely for 4 min [62]. The capillary images were registered after 20 min of rest and immediately after the PORH test. The skin capillaroscopy was performed twice during a 12-month interval.

Capillary images were archived for future analysis. Capillary coverage was calculated before (coverage_BASE_) and after the 5 min PORH test (coverage_PORH_) (Figure 1).

Capillary coverage was evaluated prior to (coverage_BASE_) and after completion of the PORH test (coverage_PORH_). The reactivity of skin microcirculation was also determined by calculating the difference in surface area covered by capillaries (Δcoverage) before and after the PORH test (Δcoverage_PB_). In addition, relative values were calculated as the ratio of Δcoverage to the corresponding pre-test coverage value (R_coverage). All data obtained during the repeat examination have the bottom index “12”.

Additionally, we have previously studied the reproducibility of capillaroscopy and have described our approach in detail in a separate report [33]. The repeatability of capillaroscopy parameters was evaluated using the intraclass correlation coefficient (ICC). The obtained ICC of 0.63 indicates good reproducibility [63], where sections of capillaroscopic images correspond to identical areas of vasculature selected during two separate examinations. This seems to be in line with the results obtained in other studies. Previous assessments of pathological capillary density via video capillaroscopy yielded similar reproducibility rates among healthy individuals as well as patients with Raynaud’s phenomenon, with values of 0.71 and 0.67, respectively. Smith et al. [64] also evaluated the reproducibility of capillary identification with the use of videocapillaroscopy in patients with systemic scleroderma. The results ranged from ICC = 0.64 for branching capillaries to ICC = 0.96 for non-vascularized fields. Similarly, Hudson et al. [65] obtained comparable values (ICC 0.72–0.84) for the repeatability assessment of capillary density using the same images or images taken at the same time of day. Ijzerman et al. [66] performed capillaroscopic studies on a group of nine subjects over two sessions to determine the coefficient of variation for capillary density assessment, yielding a result of 8.3 ± 5.9%.

### 2.3. Laboratory Analyses

Blood samples were collected to assess the levels of HbA1c, CRP, total cholesterol, high-density lipoprotein cholesterol (HDL-C), low-density lipoprotein cholesterol (LDL-C), triglycerides (TG), TSH, fT4, homocysteine, serum creatinine, TNF-α, VEGF and IL-35. TNF-α, VEGF and IL-35 were measured with the use of the enzyme-linked immunoassay ELISA (R D Systems, Minneapolis, MN, USA) according to the manufacturer’s protocol. Mean glycated hemoglobin values in the first and second year of disease duration were determined as the average of 4 measurements taken every three months in the same laboratory.

### 2.4. Statistical Analysis

All statistical analyses of the data obtained were performed using STATISTICA version 13.1 statistical software from StatSoft Inc., Tulsa, Oklahoma, USA; license CSM GUMed JPZP5077539317AR-H.

The distribution of the variables was assessed using the Shapiro–Wilk test. In the absence of a normal distribution of the study variables, their values were compared using the non-parametric Mann–Whitney *U* test. Relationships between variables were tested using Spearman’s rank correlation and *Chi*^2^ test with Yates correction where appropriate. For group changes, the Wilcoxon test was used. The *chi*-square test was used to compare gender proportions, medication use and hypoglycemic episodes. A significance level of *p* < 0.05 was considered statistically significant.

## 3. Results

Our results demonstrated that coverage_BASE_ for the entire study group correlated significantly with coverage_PORH_ (Figure 2). Following the first capillaroscopic examination, patients were divided into two subgroups according to the median coverage at baseline (coverage_BASE_) (Table 1). Subgroup A consisted of patients whose coverage_BASE_ was under the median value for the whole group, and subgroup B consisted of patients whose coverage_BASE_ value was above the median value for the whole group. Coverage values were then obtained for each subgroup at baseline and after one year. The values were then analyzed and compared between subgroups.

### 3.1. Subgroup Comparison

There was no significant difference between age, onset of diabetes or disease duration in the studied subgroups. The patients did not differ with respect to their BMI, number of hypoglycemic episodes or dose and method of insulin administration.

HbA1c levels at disease onset as well as levels obtained within the first and second year of disease duration were significantly higher in subgroup A when compared to subgroup B. In contrast, the subgroups of patients with diabetes did not differ in HbA1 during the present study (Table 1).

The studied subgroups did not differ with regard to the total cholesterol level and its fractions, free T4 levels, creatinine levels, albuminuria and CRP. The only significant difference that was found was between TSH levels, which were higher in subgroup A. However, it should be noted that all subjects were in euthyroid (Table 2).

Upon comparing the biochemical parameters obtained at baseline to those values obtained after one year, we found a significant increase in albuminuria in the primary study group as well as in subgroup B (*p =* 0.05). However, despite this increase, the values remained within the normal range.

Additionally, there was a significant decrease in LDL cholesterol in subgroup A and a significant increase in triglycerides in subgroup B (Table 2).

In subgroup A, distinguished by lower baseline capillary coverage, we found significantly higher levels of TNF-α, VEGF and the TNF-α/IL-35 ratio when compared to subgroup B. On the other hand, IL-35 levels tended to be lower in subgroup B (*p =* 0.08) (Table 2).

We found that subgroups divided based on baseline capillary coverage demonstrated significant differences in coverage_PORH_ as well. We also found a significant increase in baseline coverage after one year (coverage_BASE_12_) (*p =* 0.03) in subgroup A, whereas there was a significant decrease in coverage after the PORH test in subgroup B (coverage_PORH___12_) (*p =* 0.02).

We have analyzed the change between the coverage_BASE_ values obtained at the initial examination and the values obtained after 12 months (Δcoverage_BASE_). We also evaluated the change between coverage_PORH_ values during the initial exam and after 12 months (Δcoverage_PORH_).

We found that both Δcoverage_BASE_ and Δcoverage_PORH_ changed significantly over the course of the year.

The other microcirculation parameters did not differ between the two subgroups, nor did they change significantly over the course of the year (Table 3).

### 3.2. Correlations between Variables

Within the primary study group of diabetic patients, we found a significant inverse correlation between coverage_PORH_ and HbA1c values measured at the onset of the disease (r = −0.38, *p =* 0.01) and at 1 year after diagnosis (r = −0.43, *p =* 0.003). Δcoverage_PB_ and R_coverage (relative reactivity) correlated significantly with HbA1c during the initial examination (r = 0.41 *p =* 0.005; r = 0.41, *p =* 0.005, respectively) and HbA1c at one year after (r = 0.35, *p =* 0.02; r = 0.34, *p =* 0.02, respectively). Other parameters of skin microcirculation did not correlate with HbA1c.

There was no evidence that patient age, age at onset or duration of diabetes had any effect on any of the capillaroscopy parameters. No significant correlation was found between the insulin dose or method of administration and capillaroscopy parameters or cytokine levels.

The TNF-α/IL-35 ratio correlated significantly with HbA1c at the time of diabetes onset (r = 0.52, *p* < 0.001), as well as with HbA1c after 1 year (r = 0.38, *p =* 0.009). However, there was no significant correlation between the TNF-α/IL 35 ratio and HbA1c at baseline and the following year. We have noticed similar correlations between VEGF and HbA1c at the onset of the disease and in the 1st and 2nd year of the disease (r = 0.57, *p* < 0.001; r = 0.54, *p* < 0.001 and r = 0.31, *p =* 0.04; respectively). TNF-α, on the other hand, was only correlated with HbA1c levels at disease onset (r = 0.52, *p* < 0.001) and during the 1st year of the disease (r = 0.38, *p =* 0.009). With regards to IL-35, we found a negative correlation with Hba1c at disease onset (r = −0.42, *p =* 0.004) and after 1 year of the disease (r = −0.32, *p =* 0.03).

There was a significant negative correlation between the TNF-α/IL-35 ratio, coverage_BASE_ (r = −0.32, *p =* 0.03) and coverage_PORH_ (r = −0.52, *p* < 0.001). The relationship between TNF-α and other microcirculation parameters was also negative (Figure 3 and Figure 4). No significant correlations were observed between capillaroscopy parameters and IL-35 or VEGF levels (Figure 3 and Figure 4).

In the studied group of children and adolescents with diabetes, significant positive correlations were found between VEGF and TNF-α (r = 0.52, *p* < 0.001), TNF-α/IL-35 ratio (r = 0.59, *p* < 0.001). On the other hand, we noted a negative correlation between VEGF and IL-35 (r= −0.41, *p =* 0.004). There were no significant correlations between IL-35 and TNF-α.

## 4. Discussion

In a study of 123 young patients with type 1 diabetes, Zorena et al. assessed IL-12 and TNF-α levels and their association with the presence of microangiopathy. They concluded that an underlying imbalance between the pro- and anti-angiogenic function of these cytokines drives the mechanism by which TNF-α and IL-12 shape the course of the disease [67]. In an age-matched group of patients with diabetes, as well as some with diabetes and retinopathy, they showed that TNF-α was the most important predictor of vision damage. They postulate that early introduction of TNF-α antagonists into treatment regiments of young patients with type 1 diabetes, who appear to have high serum cytokine activity, may prevent the development of diabetic retinopathy [22]. This seems in line with the data presented by Yokoi et al., who demonstrated that TNF-α plays a role in the development of diabetic retinopathy through the activation of pro-inflammatory cytokines in response to hyperglycemia [68]. Some of the diabetic patients in our study were found to have a smaller surface area of vascular coverage.

VEGF is a predictor and risk factor for microalbuminuria and early diabetic nephropathy in adolescents and young adults with childhood diabetes [69]. In a study of 196 young diabetic patients aged 2–24, with diabetes onset before the age of 12 and the disease duration of at least 2 years, Chiarelli et al. [70] showed that serum VEGF concentrations were increased in prepubertal and pubertal children with diabetes. They also demonstrated that glycemic control affects serum VEGF levels, and that the severity of microvascular complications was associated with a significant increase in serum VEGF in this group of patients. Studies have shown that in patients with elevated levels of anti-inflammatory cytokines, namely L-35, the pancreatic β-cell secretory function is preserved for a longer period of time, reducing the risk of hypoglycemia and diabetic complications [71].

In the primary study group of children and adolescents with diabetes, we have found significant positive correlations between VEGF and TNF-a, as well as VEGF and the TNFa/IL-35 ratio. Meanwhile, a significant negative correlation was noted between VEGF and IL-35 levels. We found no significant correlations between IL-35 and TNF-a.

IL-35 was first identified by Collison in 2007 and, as mentioned in the introduction, belongs to the IL-12 family [66,72]. It is released from a wide range of immune system cells, including Tregs, IL-35-producing regulatory T cells (iTr35), regulatory B cells (Bregs) and tolerogenic DCs (DCs with immunoregulatory properties) [73].

The significance of IL-35 has been noted in multiple inflammatory diseases affecting the digestive, nervous, bone and respiratory systems [74]. IL-35 levels have been found to be significantly reduced in patients with ulcerative colitis and multiple sclerosis. IL-35 has also been shown to prevent the development of autoimmune diabetes [74,75,76]. This cytokine is highly unconventional given the number and types of receptors it utilizes, as well as the various pathways it induces [72,73]. IL-35 receptors were first identified in T-cells, which constitute an IL-12Rβ2/gp130 heterodimer or homodimer of either chain [72]. Although IL-12RB2 and GP130 homodimers may play a part in IL-35—mediated immunosuppressive signaling, its maximal function is elicited by the IL12RB2/gp130 heterodimeric receptor. It triggers signaling pathways from the signal transducer and activator of transcription 1 (STAT1) and it is a signal transducer and activator of the transcription 4 (STAT4) switch [72,73].

Nordwall et al. [77] showed that a weighted average of HbA1c measured levels, obtained over a long period of time starting at the time of diagnosis, is significantly associated with the development of complications in type 1 diabetes. In fact, maintaining HbA1c levels below 7.6% appears to prevent proliferative retinopathy and macroalbuminuria for up to 20 years after diagnosis.

The authors of an Italian study found that among the clinical, metabolic, immunological and biochemical factors assessed, only HbA1c at the onset of diabetes was predictive of the development of microangiopathy in type 1 diabetes patients [78]. The phenomenon of metabolic memory [78] described by these authors is consistent with our results. In our cohort of diabetic patients, only HbA1c at disease onset and in the first year of disease duration allowed us to distinguish patients with impaired skin microcirculation. In turn, glycemic control later in the course of the disease did not appear to be a significant predictor of microvascular function.

Several non-invasive methods are currently available for the assessment of skin microcirculation. Capillaroscopy and videocapillaroscopy, laser Doppler flowmetry, thermography and transcutaneous measurement of oxygen tension (tcpO2) are all of great clinical value [61,79]. Capillaroscopy is a widely accepted clinical management component in patients with Raynaud’s phenomenon [49,50], systemic scleroderma [49,50,51,52], undifferentiated connective tissue disease [53], Behçet’s disease [54], dermatomyositis or Sjögren’s syndrome [55]. The pattern of capillaroscopic changes specific to each of these disease processes has been identified. A strong relationship between these changes and the severity of clinical illness has also been demonstrated [50,52,56,57]. Cutaneous microcirculation abnormalities are observed in patients with cardiac syndrome X [80] and healthy individuals at high cardiovascular risk [66]. Using capillaroscopy and an ischaemic test, Ijzerman et al. [66] demonstrated a negative correlation between individual Framingham cardiovascular risk and the percentage of capillary recruitment. They suggested that microvascular function could be monitored for the effects of various cardiovascular risk factors by analyzing skin microcirculation.

Regardless of additional underlying conditions or risk factors, individuals with diabetes have been shown to have evidence of capillary remodeling and thinning. This has been demonstrated by a histological examination of muscle tissue obtained via biopsy. In fact, Jörneskog et al. [26] showed that lower extremity microcirculation of diabetic patients was significantly impaired compared to controls, irrespective of the presence of end-organ complications. Tibrica et al. [27] demonstrated impaired cutaneous microvascular function in patients with type 1 diabetes that was detectable for an average of 9.5 years in the absence of end-organ complications. Abi-Chahina et al. [81] compared the capillary area at baseline and after the PORH test between diabetic and healthy controls. They showed that the change in the area covered by capillaries following the PORH test was significantly smaller in the diabetic group. On the other hand, Schalkwijk et al. [82] demonstrated that the phenomenon of capillary thinning in patients with type 1 diabetes occurs only in patients with late complications of the disease.

Despite some inconsistencies in the available literature, certain changes in cutaneous microcirculation of diabetic patients with diabetes, including capillary thinning, have been widely observed.

It is noteworthy that among the subgroups divided based on coverage_BASE_, there were no significant differences in terms of patient age, age at disease onset or diabetes duration or current glycemic control. In contrast, Kuryliszyn [83], Tooke [84] and Tibrica [85] demonstrated a correlation between diabetes duration and capillaroscopic imaging, whereas Gasser [34] did not.

Tehrani et al. [86] also found that impaired skin microvascular reactivity could indicate clinical microangiopathy in type 1 diabetes. However, our results show that reduced capillary density and high levels of TNF-α and VEGF can precede the onset of microangiopathic complications in type 1 diabetes and may help predict abnormal microvascular reactivity. Kuryliszyn et al. showed that qualitative abnormalities in capillaroscopy reflect the progression of diabetic microangiopathy and are associated with increased VEGF levels [19]. In their study [60], they used capillaroscopy to examine skin microcirculation in patients with long-term diabetes aged 37.60 ± 12.83 years. Their results indicated that an increase in the severity of microvascular dysfunction is associated with microangiopathic complications and higher levels of selectin E and IL-18.

Our study does have certain limitations. Firstly, immunological testing was performed only at baseline and was not repeated at follow-up. Another limitation is that the number of subjects was relatively small. Although there was no control group, we do not think it was necessary to include one as the primary aim of the study was to compare values between diabetic patients at different points in time.

To the best of our knowledge, the relationship between the skin microcirculation function in children and adolescents with type 1 diabetes and the VEGF, TNF-α and IL-35 levels has not been previously described in the literature.

As we have shown in our previous study with adult diabetic patients, different vascular beds may be affected asynchronously [33] and there are many factors that are involved in the process of microvascular dysfunction. We believe that changes in skin microcirculation precede the development of clinically apparent angiopathy. This was supported by the data obtained in our current study. Our analysis demonstrated that subgroups divided based on coverage_BASE_ differed significantly in terms of HbA1c levels at the start of the study, as well as mean HbA1c levels at one and two years following diagnosis. Moreover, they had a significantly different ratio of pro- and anti-inflammatory cytokines. Our data provides further evidence that changes in skin microcirculation precede the development of other forms of angiopathies.

It appears that capillaroscopy with PORH, a relatively simple, non-invasive test, could add significant value for the early identification of microvascular dysfunction in diabetic patients. It has the potential to be included alongside the variety of recommended screening tests for type 1 diabetes, such as fundoscopic examination. However, further research and a larger number of patients are needed to determine the potential timing of such examinations in relation to both age at onset and diabetes duration. Moreover, it is possible that parallel assessment of cytokine levels may provide valuable insight into the pathogenesis and dynamics of microvascular complications in patients with type 1 diabetes.

## 5. Conclusions

Our results support the hypothesis that diabetic cutaneous microangiopathy is influenced by metabolic memory [78] and an imbalance between pro- and anti-inflammatory cytokines.

## Figures and Tables

**Figure 1 biomedicines-11-02857-f001:**
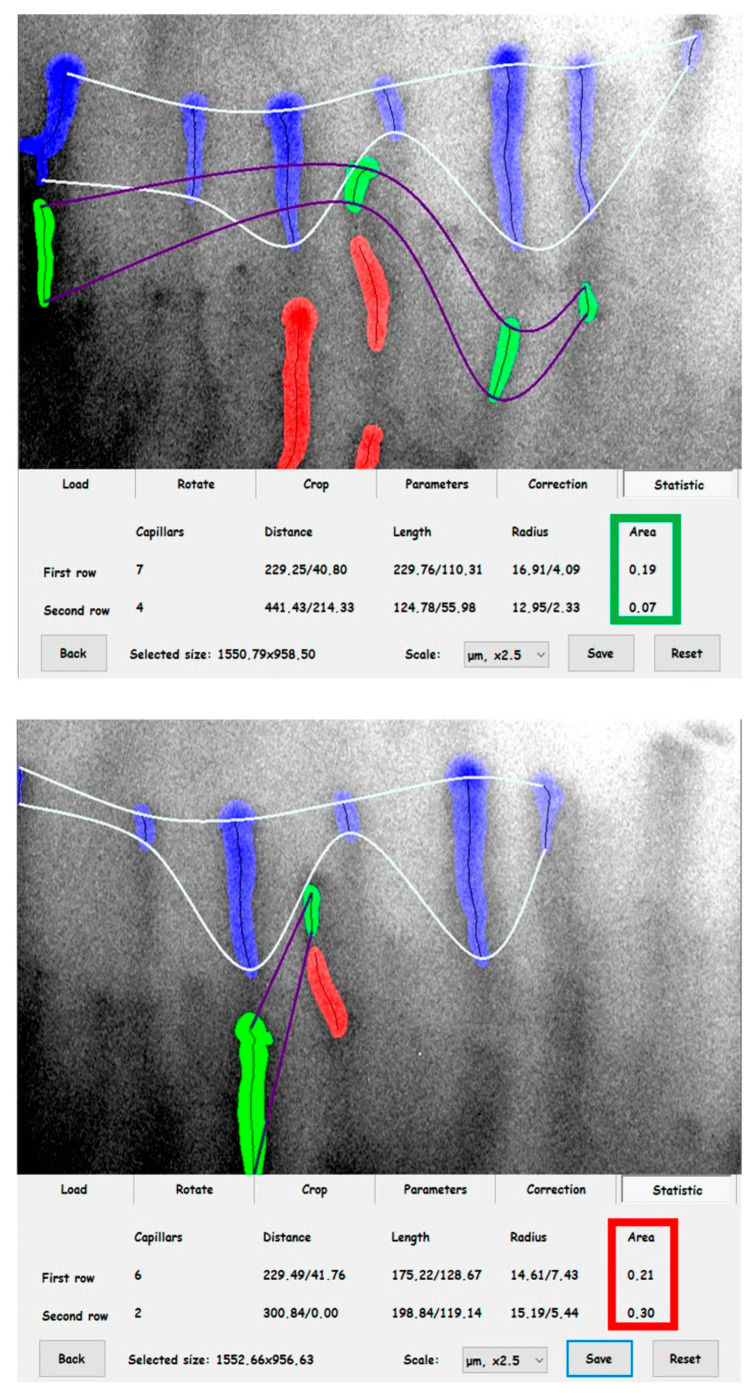
The images demonstrate skin microcirculation of a patient in the PORH test (coverage_BASE_, top image) and after test completion (coverage_PORH_, bottom image). The coverage_BASE_ values have been marked in a green frame, and the red coverage_PORH_ values have been marked in a red frame [33,61].

**Figure 2 biomedicines-11-02857-f002:**
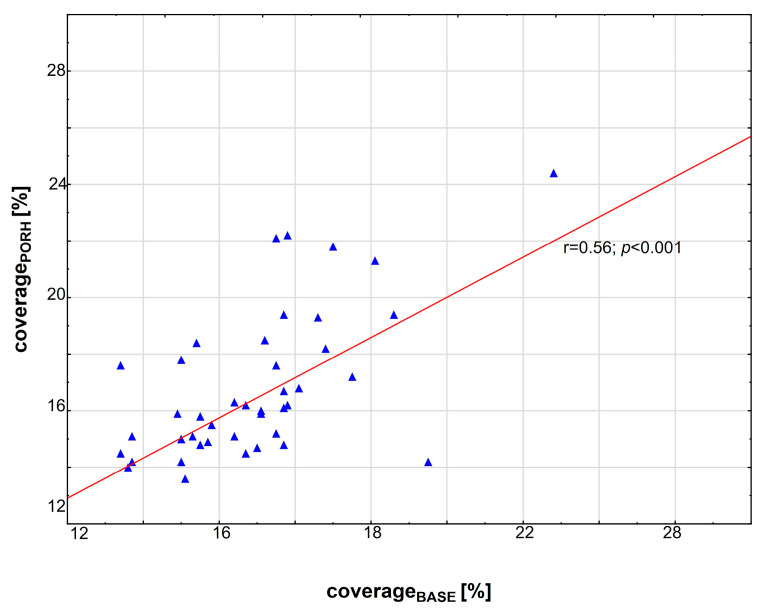
Scatterplots illustrating the relationship between coverage_BASE_ and coverage_PORH_ in the entire study group. The value of *p* < 0.05 was regarded as statistically significant. Blue triangles indicate median values.

**Figure 3 biomedicines-11-02857-f003:**
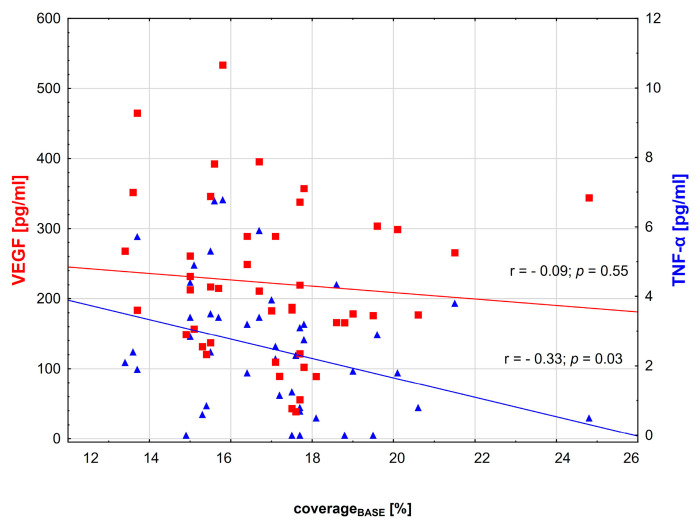
Scatterplots illustrating the relationship between plasma TNF-α, VEGF and coverage_BASE_ in the primary study group of diabetic patients. The value of *p* < 0.05 was regarded as statistically significant. Red squares and blue triangles indicate median for respective variable.

**Figure 4 biomedicines-11-02857-f004:**
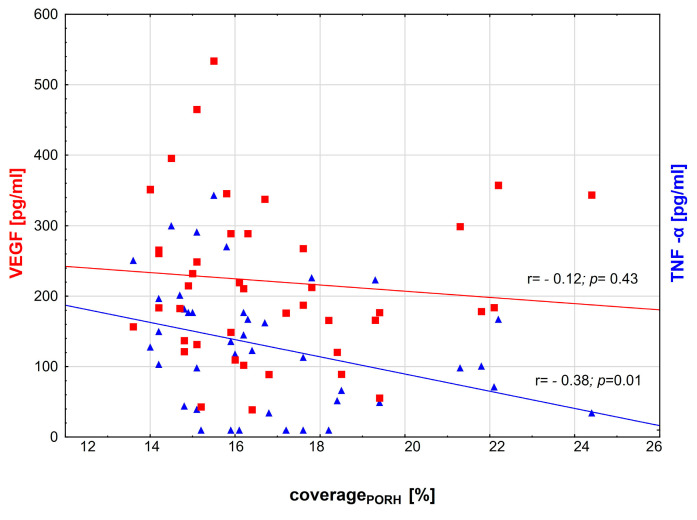
Scatterplots illustrating the relationship between plasma TNF-α, VEGF and coverage_PORH_ in the primary study group of diabetic patients. The value of *p* < 0.05 was regarded as statistically significant. Red squares and blue triangles indicate median for respective variable.

**Table 1 biomedicines-11-02857-t001:** Characteristics of all diabetic patients enrolled in the study as well as both of the subgroups divided according to median of coverage at baseline.

Characteristics	Diabetic Patients *n* = 46	Diabetic Patients Subgroups According to Median Coverage_base_		*p* for between Subgroups Comparison
		Subgroup A, *n* = 23	subgroup B, *n* = 23	
	Mean ± SD/ Median (Range)	Mean ± SD/ Median (Range)	Mean ± SD/ Median (Range)	
Males, n (%)	22 (47.8)	12 (50)	11 (47.8)	1
Age [years]	15.6 ± 2.4/ 15.6 (8.4–18)	15.0 ± 2/ 15.4 (11.1–18)	15.2 ± 2.8/ 15.7 (8.4–18)	0.59
Onset of diabetes [age]	7.3 ± 3.7/ 8.2 (1.2–13.8)	7. 2 ± 3.7/ 8.4 (1.2–11.8)	7.4 ± 4/ 7.6 (1.8–13.8)	0.82
Diabetes duration [years]	7.0 ± 3.9/ 7.0 (1.7–15.9)	7.8 ± 4/ 7.6 (1.8–14.6)	7.7 ± 4 / 7.00 (1.7–15.9)	0.85
Autoimmune thyroiditis	18 (39.1)	8 (34.8)	10 (43.5)	0.54
Treatment with statins, n (%)	3 (6.5)	0	3 (13)	0.23
BMI [kg/m²]	20.4 (15–26.8)	20.2 (15.7–24.8)	20.4 (15–26.8)	0.88
HBA1c at onset of DM [%]	9.9 (6.5–16.4)	11.8 (6.6–16.4)	9.1 (6.5–14.7)	0.02
HBA1c 1st year of DM [%]	8.2 (6.2–12.7)	9.3 (6.5–12.7)	7.90 (6.2–10.5)	0.04
HBA1c 2nd year of DM [%]	7.35 (5.5–10.4)	7.8 (5.5–10)	7.00 (5.7–10.4)	0.04
HBA1c at initiation of study [%]	8.1 (5.9–13.4)	8.1 (5.9–13.4)	8.0 (6.3–11.8)	0.44
HBA1c following one year [%]	8.5 (5.9–13.9)	8.6 (5.9–13.9)	8.0 (6.3–12.2)	0.11
Insulin dose units/24 h	46 (20–100)	46 (20–90)	45 (21–100)	0.88
Treatment with pump [%]	28 (60.2)	12 (52.2)	16 (69.6)	0.22
Time of pump treatment in ratio to DM duration [%]	64 (0–100)	19 (0 -95)	75.6 (0–100)	0.18
Episodes of mild hypoglicaemia [N/last month]	10 (3–30)	10 (3.0–20)	10 (3–30)	0.80
Episodes of severe hypoglicaemia [N/last year]	0 (0–1)	0 (0–1)	0 (0–1)	0.57

The value of *p* < 0.05 was regarded as statistically significant. Abbreviations: HbA1C—glycated hemoglobin; BMI—body mass index; DM—diabetes mellitus.

**Table 2 biomedicines-11-02857-t002:** Laboratory results of all diabetic patients and subgroups divided according to median of coverage at baseline. Data at baseline and following one year.

Characteristics	Diabetic Patients	Diabetic Patients Subgroups According to Median Coverage_BASE_		*p* for between subgroups Comparison
	*n* = 46	Subgroup A, *n* = 23	Subgroup B, *n* = 23	
	Median (Range)	Median (Range)	Median (Range)	
Total cholesterol [mg/dL]				
baseline	181 (125–288)	184 (135–288)	177 (125–285)	0.44
after one year	180 (120–319)	177 (148–315)	184 (120–319)	0.59
*p* for one year comparison	0.66	0.05	0.15	
Cholesterol LDL [mg/dL]				
baseline	108 (61–192)	111 (61–188)	105 (61–192)	0.46
after one year	101 (48–200)	99 (48–140)	109 (70–200)	0.29
*p* for one year comparison	0.19	0.005	0.46	
Cholesterol HDL [mg/dL]				
baseline	55 (38–90)	59 (42–90)	54 (38–83)	0.77
after one year	54 (31–82)	53 (31–81)	54 (36–82)	0.72
*p* for one year comparison	0.68	0.16	0.52	
Triglycerides [mg/dL]				
baseline	72 (34–274)	78 (38–249)	65 (34–274)	0.10
after one year	78 (34–583)	91 (34–408)	70 (50–583)	0.22
*p* for one year comparison	0.06	0.60	0.02	
Serum creatinine [mg/dL]				
baseline	0.68 (0.45–0.95)	0.69 (0.5–0.95)	0.65 (0.45–0.95)	0.98
after one year	0.67 (0.44–0.93)	0.67 (0.44–0.93)	0.62 (0.45–0.91)	0.75
*p* for one year comparison	0.03	0.01	0.53	
Albuminuria [mg/dL]				
baseline	6.9 (2.5–28)	8.2 (2.5–24.4)	6 (2.5–28)	0.60
after one year	9.79 (5–27.74)	10 (5–25)	9 (5–27.74)	0.81
*p* for one year comparison	0.01	0.14	0.05	
TSH [mIU/L]				
baseline	1.82 (0.57–5.1)	1.6 (0.96–3.9)	2.1 (0.57–5.1)	0.15
after one year	1.77 (0.11–5.2)	1.58 (0.1–3.4)	2 (1.1–5.2)	0.02
*p* for one year comparison	0.44	0.20	0.98	
fT4 [pmol/L]				
baseline	12.5 (9.0–15)	12.5 (9–14.6)	12.5 (9.33–15)	0.52
after one year	13 (9.0–16)	13 (9–16)	13 (10.8–15.3)	0.68
*p* for one year comparison	0.03	0.32	0.08	
C-reactive protein [mg/L]	0.41 (0.1–4.8)	0.37 (0.1–4.8)	0.44 (0.1–4.3)	0.64
TNF-α [pg/mL]	2.4 (0–6.78)	3.4 (0–6.8)	1.25 (0–4.3)	0.001
IL-35 [ng/mL]	4.85 (1.2–22.67)	3.9 (1.2–22.4)	6.8 (1.8–22.7)	0.08
ratio TNF-α/IL-35	0.44 (0–3.44)	0.89 (0–3.4)	0.18 (0–1.1)	0.001
VEGF [pg/mL]	211.75 (38.8–533.73)	232.2 (120.4–533.7)	176.8 (38.81–357.4)	0.03

The value of *p* < 0.05 was regarded as statistically significant. Abbreviations: HbA1C: glycated hemoglobin; TSH: Thyroid-stimulating hormone; fT4: free thyroxine; LDL—low density lipoproteins; HDL—high density lipoproteins; TNF-α—tumor necrosis factor; IL-35—interleukin 35; VEGF—vascular endothelium grow factor.

**Table 3 biomedicines-11-02857-t003:** Characteristics of skin microcirculation of the primary study group of diabetic patients and subgroups divided according to median coverage at baseline. Data at baseline and following one year.

Characteristics	Diabetic Patients	Diabetic Patients Subgroups According to Coverage_BASE_ Median		*p* for between
	*n* = 46	Subgroup A *n* = 23	Subgroup B *n* = 23	Subgroups Comparison
	Median (Range)	Median (Range)	Median (range)	
coverage before PORH test [ %] (coverage_BASE_)				
baseline	17.1 (12.7–24.8)	15.4 (12.7–17)	17.8 (17.1–24.8)	---
after one year	18 (12–23)	17(12–22)	18 (15–23)	0.33
*p* for one year comparison	0.19	0.03	0.54	
difference between Δcoverage_BASE_12_ and coverage_BASE_ [%] (Δcoverega_BASE_)				
	0.7 ((−8.8)7.6)	2 ((−3.6)–7.6)	−0.1 ((−9.4)–7.5)	0.03
coverage after PORH test [%] (coverage_PORH_)				
baseline	16 (10.4–24.4)	15 (10.4–18.4)	17.2 (11.5–24.4)	<0.001
after one year	16 (11–24)	17 (11–24)	16 (11–20)	0.56
*p* for one year comparison	0.88	0.07	0.02	
difference between Δcoverage_PORH_12_ and coverage_PORH_ [%] (Δcoverega_PORH_)				
	−0.3 ((−9.4)–12.6)	2.3 ((−4.5)–12.6))	−1.2 ((−9.4)–7.5)	0.005
capillary reactivity [%] Δcoverage between coverage after and before PORH test [%] (Δcoverage_BP_)				
baseline	−0.7 ((−8.1)–4.6)	−0.3 ((−4.1)–4.2)	−1.1 ((−8.1)–4.6)	0.57
after one year	−1.0 ((−1.0)–5.0)	−1 ((−6)–5)	−1 ((−10)–2)	0.10
*p* for one year comparison	0.18	0.72	0.11	
(the ratio of Δcoverage between coverage after and before PORH test, and coverage before PORH test) (R_coverage)				
baseline	−3.9 ((−41.3)–31.3)	−1.9 ((−26.5)–31.3)	−5.8 ((−41.3)–26.3)	0.63
after one year	−4.5 ((−43)–44)	−1.0 ((−28)–44)	−7.0 ((−43)–16)	0.16
*p* for one year comparison	0.30	0.97	0.12	
ratio between capillary reactivity after one year to capillary reactivity at baseline				
	0 ((−19)–18)	0.5 ((−18)–19)	0 ((−12)–4)	0.44

The value of *p* < 0.05 was regarded as statistically significant.

## Data Availability

The research data can be requested from the first author.

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
