# Peer review of "The Relationship between TNF-a, IL-35, VEGF and Cutaneous Microvascular Dysfunction in Young Patients with Uncomplicated Type 1 Diabetes"

_biomedicines, 2023, doi:10.3390/biomedicines11102857_

Round 1

Reviewer 1 Report

This paper provides results on potential relationship between immunological markers and skin microcirculation dysfunction in young patients with type 1 diabetes. Although the study provides some interesting details, the authors still need to clarify some particularly methodological issues.

What bothers me most is the division of respondents into subgroups, for which there are really no criteria or basis in the literature. Indeed, in the title itself, the authors refer to microvascular dysfunction, but in fact, nowhere in the paper are the criteria for what counts as microvascular dysfunction mentioned. So it turns out that the median of a randomly selected population of 46 patients with DM1 was taken as the cutoff point to declare something as good function or dysfunction. In my opinion, the authors should really reflect on or provide evidence for this classification of respondents.

Moreover, there is also a question why second measurement was included in study design, as it was not complete follow-up to the first one?

Special attention should be given to the reproducibility and mechanisms involved in the microcirculation function test assessed by videocapillaroscopy.

Introduction

Line 52-53 Is this mean that there is negative correlation between IL-35 level and left ventricular ejection fraction?

Lines 54-59 What are anti-inflammatory effects of IL-35 in sepsis?

In the introduction, a brief review supported by references should be added on capillary videocapillaroscopy as a method for the assessment of microvascular reactivity; which pathways are involved in the microcirculation response to vascular occlusion, i.e. which mechanisms mediate post-occlusive reactive hyperemia.

Materials and Methods

Please explain if the duration of diabetes in an individual correlated with microvascular reactivity, i.e. influenced the division of subjects into groups.

What was the longest duration of the DM1 in an individual? Why did you decide to monitor HbA1c in the first two years of the disease?

Please provide rationale for study follow-up 1 year after initial testing, especially since pro- and anti-inflammatory mediators were not assessed at this occasion.

Results

Explain in detail why the groups of respondents were divided in subgroups in this manner? Why was the median measurement taken as a cut-off for dividing respondents into subgroups? What are the day-to-day and inter-subject reproducibility of post occlusive reactive hyperaemia (PORH) measured by capillaroscopy?

3 patients in subgroup B were taking statin therapy? Treatment of hyperlipidemia may present a confounding factor as it may also affect microvascular reactivity? Why didn't you decide to exclude those patients?

Lines 192-193 Is this correct?

Lines 206-209 Please explain this sentence.

Lines 248-249 What is rationale to correlate currently measured IL-35 with HbA1c at the beginning of the disease? Same goes for VEGF and TNF-alpha.

Discussion

Discussion section should start with the most important results of the study and not by repeating the aims and hypothesis.

Extensive editing of English language required.

Author Response

This paper provides results on potential relationship between immunological markers and skin microcirculation dysfunction in young patients with type 1 diabetes. Although the study provides some interesting details, the authors still need to clarify some particularly methodological issues.
What bothers me most is the division of respondents into subgroups, for which there are really no criteria or basis in the literature. Indeed, in the title itself, the authors refer to microvascular dysfunction, but in fact, nowhere in the paper are the criteria for what counts as microvascular dysfunction mentioned. So it turns out that the median of a randomly selected population of 46 patients with DM1 was taken as the cutoff point to declare something as good function or dysfunction. In my opinion, the authors should really reflect on or provide evidence for this classification of respondents.

Answer: We would like to thank the Reviewer for raising the issue of subgroups creation. In the presented research, we aimed to examine the microcirculatory response to the PORH stimulus in children with diabetes mellitus, comparing those
with higher pre-test coverage values to those with lower coverage values. To achieve this, we divided the whole group in respect to median pre-PORH coverage, without implying that those with coverage below the median exhibited
microvascular dysfunction.
Moreover, there is also a question why second measurement was included in study design, as it was not complete follow-up to the first one?
Special attention should be given to the reproducibility and mechanisms involved in the microcirculation function test assessed by videocapillaroscopy.

Answer: We would like to precise that in presented report we didn't use videocapillaroscopy, but a stereomicroscope technique with spot lighting, as detailed in our paper.
We agree with the Reviewer’s opinion that reproducibility is the issue of great importance. Therefore we added following
information to the section 2.2.
“Additionally, we have previously studied reproducibility of capillaroscopy and have de-scribed our approach in detail in a separate report [33]. The repeatability of capillaroscopy parameters was evaluated using the intraclass correlation
coefficient (ICC). The obtained ICC of 0.63 indicates good reproducibility [63], where sections of capillaroscopic images correspond to identical areas of vasculature selected during two separate examinations. This seems to be in line with results obtained in other studies. Previous assessments of pathological capillary density via video capillaroscopy yielded similar reproducibility rates among healthy individuals as well as patients with Raynaud’s phenomenon, with values of 0.71 and 0.67, respectively. Smith et al [64] also evaluated reproducibility of capillary identification with the use of videocapillaroscopy in patients with systemic scleroderma. The results ranged from ICC=0.64 for branching capillaries to ICC=0.96 for non-vascularized fields. Similarly, Hudson et al [65] obtained comparable values (ICC 0.72-0.84) for repeatability assessment of capillary density using the same images or im-ages taken at the same time of day. Ijzerman
et al [66] performed capillaroscopic studies on a group of nine subjects over two sessions to determine the coefficient of variation for capillary density assessment, yielding a result of 8.3±5.9%.”
Introduction
Line 52-53 Is this mean that there is negative correlation between IL-35 level and left ventricular ejection fraction?

Answer: The data presented in lines 52-53 exactly recall the result presented in report by Lin et al. linking reduced IL35
values with LVEF. Meaning that lower IL-35 values correlate with lower ejection fraction [8, Figure 3].
Lines 54-59 What are anti-inflammatory effects of IL-35 in sepsis?

Answer: Thank you for your question regarding this issue. We hope that the following paragraph we added will be sufficient explanation regarding relation between IL-35 and sepsis.
„ Interleukin-35 inhibits activation of vascular endothelial cells by blocking MAPK-AP1-mediated VCAM-1 expression during acute inflammation induced by lipopolysaccharides. This leads to inhibition of the acute vascular endothelial
response.”
In the introduction, a brief review supported by references should be added on capillary videocapillaroscopy as a method for the assessment of microvascular reactivity;
We Reviewer’s opinion and do hope that the following sentences will sufficiently extent the methodological issues
“Capillaroscopy or videocapillaroscopy is a method commonly used by researchers in the assessment of cutaneous microvasculature and evaluation of endothelial function. Researchers have utilized capillaroscopy when studying various
disease entities, namely peripheral vascular disease [30], hypertension [31], chronic kidney disease [32], type 1 diabetes [27, 33-37], type 2 diabetes [38], obesity [39-41] or post bariatric procedures [42]. Microcirculation reactivity has also been
investigated in patients suffering from the infective endocarditis [43] or ischemic, no obstructive coronary artery disease
[44].”
Which pathways are involved in the microcirculation response to vascular occlusion, i.e. which mechanisms mediate post-occlusive reactive hyperemia.

Answer: We would like to thank the Reviewer for the chance to expand the introduction section with the description of
PORH test.
“For our study on pediatric patients we chose capillaroscopy with employed non-selective stimuli such as post-occlusive reactive hyperemia test. Reactivity tests, such as the venous occlusion test and arterial post-occlussive reactive hyperemia (PORH), have been suggested to improve capillary recruitment. They allow to determine the total maxi-mal capillary density with high reproducibility [45]. PORH refers to the increase from baseline in cutaneous blood flow following a brief period of arterial occlusion [46]. A wide range of brachial artery occlusion times has been described, with studies implementing times anywhere from 1 minute to 15 minutes. A positive correlation has been described between the post-occlusive hyperemic response and the duration of arterial occlusion [46]. PORH is a widely used test for the evaluation of microcirculation, with the main underlying mechanism being shear stress and its effects [47].
Microcirculatory function, specifically the degree of capillary recruitment during occlude reactive congestion, is associated with endothelium-dependent vasodilation at the pre-capillary level. This is mediated by an axonal reflex [48, 49] and endothelium-derived hyper polarizing factor. The PORH test induces relaxation of the vascular muscle [50], and the release of local mediators and metabolites from the ischemic tissue [51]. It has also been demonstrated that local mediators, in particular large conductance calcium-activated potassium channels [49], play an important role in the PORH mechanism [48].”
Materials and Methods
Please explain if the duration of diabetes in an individual correlated with microvascular reactivity, i.e. influenced
the division of subjects into groups.

Answer: Thank you for your question. As previously mentioned in the "Results" section, there was no correlation found between diabetes duration and any microcirculatory parameters.
What was the longest duration of the DM1 in an individual?

Answer: The longest duration of diabetes was 15.9 years.
Why did you decide to monitor HbA1c in the first two years of the disease?

Answer: Children with type 1 diabetes are cared for in the Diabetes Clinic and the Outpatient Clinic.
They undergo testing for HbA1c levels every three months. We obtained the data on these values from the available documentation. As the phenomenon of metabolic memory is well known, it is of interest to explore whether there is a potential association between glycemic control early in the course of the and disturbances in cutaneous microcirculation
function in patients without the classic micro- and macroangiopathic complications.
Please provide rationale for study follow-up 1 year after initial testing, especially since pro- and antiinflammatory mediators were not assessed at this occasion.

Answer: That was our first idea. Unfortunately, financial limitations made the idea unfeasible. However, cytokine levels determined at baseline might have predictive value for the occurrence of microcirculatory disorders at one-year follow-up.
Results
Explain in detail why the groups of respondents were divided in subgroups in this manner? Why was the median measurement taken as a cut-off for dividing respondents into subgroups?

Answer: In the presented research, we aimed to examine the microcirculatory response to the PORH stimulus in children with diabetes mellitus, comparing those with higher pre-test coverage values to those with lower coverage values. To achieve this, we divided the group based on the median pre-PORH coverage, without implying that those with coverage below the median exhibited microvascular dysfunction
What are the day-to-day and inter-subject reproducibility of post occlusive reactive hyperaemia (PORH) measured by capillaroscopy?
The issue of day-to-day and inter-subject reproducibility of PORH was not studied by us. The assessment of repeatability we described in the answer to previous Reviewer’s comment.
3 patients in subgroup B were taking statin therapy? Treatment of hyperlipidemia may present a confounding factor as it may also affect microvascular reactivity? Why didn't you decide to exclude those patients?

Answer: We decided not to exclude patients from the analysis of lipid levels. We included the three individuals who continued statin therapy at both baseline and 1 year.
Lines 192-193 Is this correct?

Answer: Yes, it is correct.
Lines 206-209 Please explain this sentence.
Answer: We precise the commented sentence and introduced it in the form as follows:
“Upon comparing the biochemical parameters obtained at baseline to that values obtained after one year, we found a significant increase in albuminuria in the primary study group, as well as in subgroup B (p=0.05). However, despite this
increase, the values remained within normal range.”
Lines 248-249 What is rationale to correlate currently measured IL-35 with HbA1c at the beginning of the disease? Same goes for VEGF and TNF-alpha.

Answer: We hypothesized that cytokine levels at the time of testing would be associated with the degree of glycemic control during the first year of disease, a phenomenon known as metabolic memory, a major factor in the development of
vascular complications.
Discussion
Discussion section should start with the most important results of the study and not by repeating the aims and hypothesis.

Answer: Following the Reviewer's advice, the discussion section has been restructured.
Comments on the Quality of English Language

Answer: The current version of the manuscript with the corrections based on the referees' comments will be checked again by one of the authors (MW), medicine doctor affiliated with Department of Pediatrics, Northwestern University
Feinberg School of Medicine; Division of Neonatology, Ann & Robert H. Lurie Children's Hospital of Chicago.

Reviewer 2 Report

I congratulate the authors, the work is well structured, interesting and scientifically valid. I consider it very interesting given the diffusion of diabetes and the impact on the healthcare system in terms of costs.

I have only a few minor considerations.

1.       It would have been interesting to have a series of healthy control subjects, perhaps even just 15 - 20 subjects, to verify the levels of the studied cytokines (probably almost absent) compared to patients.

2.       The authors in lines 211 - 214 describe that the group of patients A (lower capillary coverage) show higher levels of TNF-alpha, VEGF and the TNF-a/IL-35 ratio compared to the subgroup B (high capillary coverage). Given that the main source of TNF are monocytes, macrophages and the main source of IL-35 are regulatory T lymphocytes and activated T lymphocytes, it would have been appropriate to include an extended phenotype among the investigations to verify the levels of regulatory T and activated T lymphocytes and their trend in the two subgroups. If data on the levels of activated T cells (CD38+, HLA-DR+, CD40L+/CD137+) or regulatory T cells (CD25+, CD127low) are available they should be added. Similarly, the levels of activated monocytes would be interesting CD68+, CD14+, HLA-DR+, CD62L+, CD18+.

3.       As with the point above, it would have been interesting to evaluate the production levels of other cytokines such as IL-10, IL-12 and IL-1beta, which are often expressed together with TNF-alpha in an inflammatory process. IL-1 beta mimics many effects of TNF while IL-10 is produced by activated macrophages, with a certain delay, after the production of TNF. it is reasonable to think that where there were higher levels of TNF, higher levels of IL-1 and perhaps also IL-10 could be found. If these dosages have been done, the data should be added to enrich the work.

4.       In the discussion section the authors describe the influence of TNF alpha and VEGF in diabetes and the link between their levels and microangiopathies such as diabetic nephropathy etc. I would ask the authors to add a few lines to underline the importance of IL-35 in diabetes and more generally in autoimmune diseases and its link with the inflammatory process and its characteristic as an anti-inflammatory cytokine. Perhaps also introducing two comments on molecular pathways (signal transduction via gp130 and IL-12RB2 and activation of STAT1 and STAT4).

Author Response

I congratulate the authors, the work is well structured, interesting and scientifically valid. I
consider it very interesting given the diffusion of diabetes and the impact on the healthcare
system in terms of costs.
We appreciate your interest in our work.
I have only a few minor considerations.
1. It would have been interesting to have a series of healthy control subjects, perhaps even
just 15 - 20 subjects, to verify the levels of the studied cytokines (probably almost absent)
compared to patients.
Answer: Due to funding constraints, we were unable to measure cytokine levels in control subjects.
Therefore, we have excluded healthy controls from the paper.
2. The authors in lines 211 - 214 describe that the group of patients A (lower capillary
coverage) show higher levels of TNF-alpha, VEGF and the TNF-a/IL-35 ratio compared to the
subgroup B (high capillary coverage). Given that the main source of TNF are monocytes,
macrophages and the main source of IL-35 are regulatory T lymphocytes and activated T
lymphocytes, it would have been appropriate to include an extended phenotype among the
investigations to verify the levels of regulatory T and activated T lymphocytes and their trend
in the two subgroups. If data on the levels of activated T cells (CD38+, HLA-DR+,
CD40L+/CD137+) or regulatory T cells (CD25+, CD127low) are available they should be added.
Similarly, the levels of activated monocytes would be interesting CD68+, CD14+, HLA-DR+,
CD62L+, CD18+.
Answer: We acknowledge the potential benefits of analysis of additional aspects outlined in the
review, which could enhance our findings and conclusions. However, in this patient group, we were
unable to conduct such assessments.
3. As with the point above, it would have been interesting to evaluate the production levels
of other cytokines such as IL-10, IL-12 and IL-1beta, which are often expressed together with
TNF-alpha in an inflammatory process. IL-1 beta mimics many effects of TNF while IL-10 is
produced by activated macrophages, with a certain delay, after the production of TNF. it is
reasonable to think that where there were higher levels of TNF, higher levels of IL-1 and
perhaps also IL-10 could be found. If these dosages have been done, the data should be added
to enrich the work.
Answer: We are indeed convinced that, as the reviewer wrote, the determination of cytokines would
have enriched the work, but unfortunately they were not done either. Another study with a significantly
enriched panel of immunologic assays is in the planning stage.
4. In the discussion section the authors describe the influence of TNF alpha and VEGF in
diabetes and the link between their levels and microangiopathies such as diabetic nephropathy
etc. I would ask the authors to add a few lines to underline the importance of IL-35 in diabetes
and more generally in autoimmune diseases and its link with the inflammatory process and its
characteristic as an anti-inflammatory cytokine. Perhaps also introducing two comments on
molecular pathways (signal transduction via gp130 and IL-12RB2 and activation of STAT1 and
STAT4).
Answer: We have added to the Discussion Section following paragraph:
“In the primary study group of children and adolescents with diabetes, we have found significant positive
correlations between VEGF and TNF-a, as well as VEGF and the TNFa/IL-35 ratio. Meanwhile, a
significant negative correlation was noted between VEGF and IL-35 levels. We found no significant
correlations between IL-35 and TNF-a.
IL-35 was first identified by Collison in 2007 and, as mentioned in the introduction, belonged to the
IL-12 family [66, 72]. It is released from a wide range of immune system cells, including Tregs, IL-35-
producing regulatory T cells (iTr35), regulatory B cells (Bregs), tolerogenic DCs (DCs with
immunoregulatory properties) [73].
The significance of IL-35 has been noted in multiple inflammatory diseases affecting the digestive,
nervous, bone and respiratory systems [74]. IL-35 levels have been found to be significantly reduced in
patients with ulcerative colitis and multiple sclerosis. IL-35 has also been shown to prevent the
development of autoimmune diabetes [74-76]. This cytokine is highly unconventional given the number
and types of receptors it utilizes, as well as the various pathways it induces [72, 73]. IL-35 receptors
were first identifying in T-cells, which constitute an IL-12Rβ2/gp130 heterodimer or homodimers of either
chain [72]. Although IL-12RB2 and GP130 homodimers may play a part in IL-35 - mediated immunosuppressive signaling, its maximal function is elicited by IL12RB2/gp130 heterodimeric receptor. It
triggers signaling pathways from the signal transducer and activator of transcription 1 (STAT1) and
signal transducer and activator of transcription 4 (STAT4) switch [72, 73].”

Round 2

Reviewer 1 Report

The authors referred to all issues and concerns, and they have made required changes in the manuscript that in this form should be accepted for publication.